Balance training in older adults enhances feedback control after perturbations

http://orcid.org/0000-0001-8662-2625 Koster Ruud A. J. 1
http://orcid.org/0000-0002-9654-7064 Alizadehsaravi Leila 1 2
http://orcid.org/0000-0002-1033-4958 Muijres Wouter 1 3
Bruijn Sjoerd M. 1
http://orcid.org/0000-0003-2312-915X Dominici Nadia 1
http://orcid.org/0000-0002-7719-5585 van Dieën Jaap H. 1 j.van.dieen@vu.nl
1 Human Movement Sciences, VU University Amsterdam , Amsterdam , Netherlands
2 Biomechanical Engineering, Delft University of Technology , Delft , Netherlands
3 Movement Sciences, KU Leuven , Leuven , Belgium
Loenneke Jeremy
Electronic publication date: 2024 Nov 25
Publication date: 2024
Volume: 12
Electronic Location ID: e18588
Received 2024 Apr 30; Accepted 2024 Nov 4
Copyright: © 2024 Koster et al.
Copyright year: 2024
Copyright holder: Koster et al.
License: This is an open access article distributed under the terms of the Creative Commons Attribution License, which permits unrestricted use, distribution, reproduction and adaptation in any medium and for any purpose provided that it is properly attributed. For attribution, the original author(s), title, publication source (PeerJ) and either DOI or URL of the article must be cited.
License URL: https://creativecommons.org/licenses/by/4.0/

Keywords: Balance training, Feedback control, Recovery, Biomechanics, Neuromuscular control, Ageing

Funding: European Research Council (ERC) 715945 Marie Skłodowska-Curie Grant 721577 VIDI Grant 016.Vidi.178.014 Dutch Organization for Scientific Research (NWO) RK and ND received funding from the European Research Council (ERC) under the European Union’s Horizon 2020 Research and Innovation program, grant number 715945–‘‘Learn2Walk’’. LA received funding from the European Union’s Horizon 2020 research and innovation programme under the Marie Skłodowska-Curie grant agreement No 721577. SMB was supported by a VIDI grant (016.Vidi.178.014) from the Dutch Organization for Scientific Research (NWO). The funders had no role in study design, data collection and analysis, decision to publish, or preparation of the manuscript.

==============================
Background

As we age, avoiding falls becomes increasingly challenging. While balance training can mitigate such challenges, the specific mechanisms through which balance control improves remains unclear.

Methods

We investigated the impact of balance training in older adults on feedback control after perturbations, focusing on kinematic balance recovery strategies and muscle synergy activation. Twenty older adults aged over 65 underwent short-term (one session) and long-term (3-weeks, 10 sessions) balance training, and their recovery from unpredictable mediolateral perturbations was assessed. Perturbations consisted of 8° rotations of a robot-controlled platform on which participants were balancing on one leg. We measured full-body 3D kinematics and activation of 15 leg and trunk muscles, from which linear and rotational kinematic balance recovery responses and muscle synergies were obtained.

Results

Our findings revealed improved balance performance after long-term training, characterized by reduced centre of mass acceleration and (rate of change of) angular momentum. Particularly during the later stage of balance recovery the use of angular momentum to correct centre of mass displacement was reduced after training, decreasing the overshoot in body orientation. Instead, more ankle torque was used to correct centre of mass displacement, but only for perturbations in medial direction. These situation and strategy specific changes indicate adaptations in feedback control. Activation of muscle synergies during balance recovery was also affected by training, specifically the synergies responsible for leg stiffness and ankle torques. Training effects on angular momentum and the leg stiffness synergy were already evident after short-term training.

Conclusion

We conclude that balance training in older adults refines feedback control through the tuning of control strategies, ultimately enhancing the ability to recover balance.

Introduction

Loss of balance poses a significant health risk, particularly in older adults, who frequently experience injurious falls (Talbot et al., 2005). The nervous system employs two control mechanisms—anticipatory and reactive control—to maintain or recover balance in spite of perturbations (Aruin, 2016). Anticipatory, or feedforward, control is based on expectations of a perturbation and aims to minimize its impact on balance by adjusting joint orientation and/or stiffness beforehand (Massion, 1992). Reactive, or feedback, control is based on sensory information derived from perturbation effects and is essential when the nervous system lacks prior knowledge of a perturbation (Nashner & Cordo, 1981; Park, Horak & Kuo, 2004). While effective feedforward control can minimize the effect of a perturbation, it may not always be energetically advantageous or even feasible to maintain balance this way, necessitating feedback control (Santos, Kanekar & Aruin, 2010a, 2010b). In reality, adequate balance control generally relies on a combination of feedforward and feedback control mechanisms. However, in older adults it is evident that both types of control are impaired. This is a result of combined degradation of sensory channels (Goble et al., 2009), the motor system (Macaluso & De Vito, 2004), and processing systems (Shumway-Cook & Woollacott, 2001). Hence, it is important to understand how both these control mechanisms contribute to balance control and how their use can be improved.

Previous studies indicated that standing balance training in older adults improved balance performance in unperturbed and perturbed standing, and even in gait (Alizadehsaravi et al., 2022a, 2022b). Furthermore, balance training can alter vestibular and proprioceptive functioning (Sihvonen, Sipilä & Era, 2004; Wiesmeier et al., 2017), as well as neuromuscular and kinematic responses after perturbations (Chvatal & Ting, 2012; Oliveira et al., 2017; Routson et al., 2013; Welch & Ting, 2008). Previously, we reported that co-contraction around the ankle was increased after training during perturbed and unperturbed balance tasks, effectively increasing joint stiffness (Alizadehsaravi et al., 2022b). This reflected an adaptation in feedforward control. However, it remains unclear whether improved balance performance after training is also attributable to altered feedback control, and therefore applicable in a wider range of circumstances.

The feedback control repertoire for balance consists of three strategies: the change in support (stepping), ankle, and angular momentum strategies (Hof, 2007). The change in support strategy aims to displace or expand the base of support beyond the projection of the centre of mass (by stepping or grabbing a handhold). It is often a last resort, reflecting poorer balance control, and is used more by older than younger adults (Afschrift et al., 2019; McIlroy & Maki, 1996; Mille et al., 2005). The ankle strategy aims to accelerate the centre of mass towards the base of support by generating ankle moments, shifting the centre of pressure (CoP), i.e., the net point of application of the ground reaction force (GRF) (Nashner & McCollum, 1985). The angular momentum strategy aims to accelerate the centre of mass towards the base of support through a horizontal GRF component generated by a change in the angular momentum of body segments relative to the centre of mass (Hof, 2007; Nashner & McCollum, 1985; van Dieën, van Leeuwen & Faber, 2015).

The ankle and angular momentum movement strategies can be differentiated by distinct kinematics and muscle activation patterns. The ankle strategy relies on rotating the body proximal of the ankle as a whole and is characterised by a distal to proximal muscle activation pattern (Horak & Nashner, 1986). In isolation, it has the line of action of the GRF run through the centre of mass (CoM) (Hof, 2007). In contrast, the angular momentum strategy rotates individual body segments in the opposing direction and has a more concurrent activation of muscles throughout the body (Horak & Nashner, 1986; van Dieën, van Leeuwen & Faber, 2015). This strategy has the GRF line of action reoriented to induce the change in angular momentum (Hof, 2007). Hence, assessment of linear and rotational whole-body kinematics as well as muscle activation allows examination of the different feedback control strategies and how they relate to training induced balance improvements.

The angular momentum strategy has been suggested to be more robust than the ankle strategy (Horak, Nashner & Diener, 1990; Horak & Nashner, 1986; Kuo & Zajac, 1993), and its use relative to the ankle strategy increases with age and magnitude of perturbations (Alcock, O’Brien & Vanicek, 2018; Amiridis, Hatzitaki & Arabatzi, 2003; Manchester et al., 1989). Older adults rely on the angular momentum strategy at a lower level of challenge than younger adults, even during unperturbed balancing (Amiridis, Hatzitaki & Arabatzi, 2003; Manchester et al., 1989). Moreover, when minimizing postural instability is prioritized over minimizing mechanical work, the angular momentum strategy is favoured over the ankle strategy (Afschrift et al., 2016). Hence, increased reliance on this strategy may help to secure robust balance control despite age-related sensory deterioration (Hsu, Chou & Woollacott, 2013; Shaffer & Harrison, 2007). However, whether training on older adults promotes use of the angular momentum strategy as a compensation or rather makes them revert to using the ankle strategy is unclear.

This study seeks to elucidate how balance training improves feedback balance control in older adults. We aimed to examine the effects of training on unipedal balance recovery after perturbations, specifically focusing on kinematics and muscle activation. We hypothesized that balance training will tune the ankle and angular momentum control strategies, improving balance recovery responses. This would suggest that training improves sensorimotor processing, allowing for a more refined feedback control mechanism.

Methods

The data collection and training program were also previously described in Alizadehsaravi et al. (2022b) and Alizadehsaravi (2021). A total of twenty healthy older adults (71.9 ± 4.1 years old) were enrolled in a balance training program, focusing on standing balance on unstable surfaces. Participants had to be at least 65 years of age. Exclusion criteria included inability to walk 3 min unassisted; sensory, orthopaedic, and cognitive impairments; and practitioners of sports involving balance components. Participant’s balance performance, kinematics, and electromyography (EMG) were measured during a unipedal balance recovery task in three sessions: at baseline (Pre), after a single training session (Post1), and after a 3-week training program (Post2). The Pre and Post1 sessions occurred on the same day. The experimental procedures received approval from the ethical review board of the Faculty of Behavioural and Movement Sciences at the Vrije Universiteit Amsterdam (VCWE-2018-171) and all participants provided written informed consent prior to participation.

Training

In the session between the Pre and Post1 recordings, the participant was trained individually for 30 min. Subsequently, a 3-week training program with three 45-min training sessions per week was completed. These sessions were completed in groups of 6–8 participants. All training was supervised by a physical therapist. We gradually increased the difficulty of exercises by reducing hand support, moving from bipedal to unipedal stance, using more unstable support surfaces, and adding perturbations such as catching and throwing a ball and reducing visual input. For a more detailed description of the training program see Material S1.

Measurements

At each of the three recording sessions, we assessed balance recovery after medio-lateral perturbations with participants in unipedal stance, on their dominant leg, on a robot-controlled platform (HapticMaster, Motek, Amsterdam, the Netherlands). Unipedal stance was selected to ensure the task was sufficiently challenging, making improvements in performance more evident, while still being a common task in everyday life. At each session participants were first familiarised with balancing unipedally on the platform: in three trials of 20 s the rotational stiffness of the platform was gradually decreased, increasing the difficulty. Subsequently, participants performed five unipedal balance trials in each of which 12 perturbations (six medial and six lateral) were induced over 60 s. Perturbations were applied in the medial and lateral direction through rotation of the platform over a sagittal axis (amplitude of 8°). It should be noted that the actual axis of rotation was located 3 cm underneath the surface of the platform. Medial perturbations were defined as the platform rotating such that the big toe moved downward (eversion) and lateral perturbations as rotations such that the big toe moved upward (inversion). This was consistent for right- and left-leg dominant participants. The direction of the perturbations as well as the interval between perturbations (3–5 s) was randomized to avoid anticipation. Throughout the trial, participants could place their other foot down on a stable surface or grab the handrails on either side of the platform to ensure safety. If this occurred more than once in a trial or lasted for over half a second, the trial was discarded and restarted after a brief rest. Between trials, there was a rest period of at least 2 min (longer if needed) to prevent fatigue. Participants were instructed to focus on a target marked by reflective tape at eye level on a black curtain. The curtain served to reduce and standardize visual information over subjects and sessions. The robot-controlled platform registered platform orientation at a sampling rate of 100 Hz. Full-body, 3D kinematic data were obtained using eight active marker clusters attached to the participant containing three markers each (thorax, pelvis, upper arms, shanks, and feet) that were tracked at a sampling rate of 50 Hz by one Optotrak camera array (Northern Digital, Waterloo, Canada) positioned posterior of the participant. Surface electromyography data were recorded from a total of 15 muscles (TMSi, Twente, The Netherlands) at a sampling rate of 2,000 Hz. Five sets of muscles around the hip and trunk were recorded bilaterally: rectus femoris (RF), adductor longus (AL), gluteus medius (GM), biceps femoris (BF), and erector spinae (ES) at the level of L2. Additionally, five muscles of the dominant leg were recorded: tibialis anterior (TA), soleus (SO), gastrocnemius lateralis (GL), peroneus longus (PL), and vastus lateralis (VL). We use the subscript D to indicate muscles on the dominant (standing) side and subscript N to indicate the non-dominant side. Ag/AgCl electrodes ( ⊘ 11, 20 mm interelectrode distance, Ambu blue sensor N, Ambu, Ballerup, Denmark) were used in a bipolar EMG configuration. Skin preparation and electrode placement was in accordance with SENIAM recommendations (Hermens et al., 2000). For a photo of the experimental setup see Material S2.

Data analysis

Kinematics

The body’s centre of mass (CoM) was estimated from the kinematics data as the mass weighted average of positions of the centres of mass of the thorax, head, pelvis, upper arms, thighs, shanks, and feet (Cappozzo et al., 1995; Kingma et al., 1996). The segment CoM positions were assumed to be at a segment specific percentage along their longitudinal axis and their masses were estimated from their lengths and circumferences using a regression equation (de Leva, 1996). For each subject, session, and perturbation direction, the average time series of centre of mass displacement (CoM [m]), velocity (vCoM [m/s]), and acceleration (aCoM [m/s2]) in the mediolateral direction were computed. The linear kinematics (vCoM and aCoM) served as indicators of performance (Yu et al., 2008). Furthermore, we calculated total body angular momentum [kg.m2/s] (with respect to its CoM), ‘relative’ body orientation (integral of the angular momentum after division by the instantaneous moment of inertia [degree]), and the rate of change in total body angular momentum (derivative of the total body angular momentum [kg.m2/s2]) as a function of time. The latter two rotational parameters, angular momentum and its derivative, served as a second indicator of performance. The angular momentum strategy is used when the rate of change in angular momentum accelerates the centre of mass towards its pre-perturbation position. Independent of this, angular momentum may be changed to regain upright body orientation. To assess how angular momentum changes were used to maintain balance, we related the direction and timing of changes in the rate of change of angular momentum to the CoM position and body orientation.

Muscle activity

EMG data were high-pass (30 Hz, bidirectional, 3rd order Butterworth) and notch filtered (50 Hz and its harmonics up to the Nyquist frequency, ±0.5 Hz bandwidth, bidirectional, 4th order Butterworth). The filtered data were then Hilbert rectified, and low-pass filtered (5 Hz, bidirectional, 2nd order Butterworth) to obtain EMG envelopes (Boonstra & Breakspear, 2012). The EMG envelopes were downsampled to the sampling rate of the balance platform (100 Hz) to speed up calculations. For every participant each rectified EMG channel was normalized to its mean value over all five trials per session. Subsequently, [−0.5, 2.5] s windows around perturbation onset were extracted. The average EMG response to the perturbation in each direction was determined per session and participant. From this, the mean baseline (−0.5 s before perturbation) activity for each muscle was subtracted to focus on changes in activation after the perturbation. These average EMG responses from all recording sessions, participants, and perturbation directions were concatenated.

Similar to the balance control strategies, muscle activation was also investigated from a central control perspective. The control of balance has been demonstrated to have a modular component (Chvatal & Ting, 2013; Nashner & McCollum, 1985; Torres-Oviedo & Ting, 2007, 2010; Wojtara et al., 2014; Zandvoort et al., 2019). As such, the dimensionality of the EMG was reduced by decomposing the concatenated EMG into synergies, each consisting of a temporal activation profile and a spatial distribution over the muscles (weighting factors). This was achieved by means of principal component analysis (PCA). The first n components that, combined, explained at least 70% of the total variance of the concatenated EMG were extracted. The resulting temporal activation profiles were split up per perturbation and averaged to obtain participant, session, and direction specific mean synergy activation responses.

The use of PCA was preferable over the conventional non-negative matrix factorization (NNMF) approach as the validity of NNMF can be limited when a subset of muscles shows mostly tonic activity. Due to its non-negativity constraint, the activation patterns of (at least) two synergies will tend to be reciprocal to explain the muscles with non-zero flat line activation patterns. As PCA does not have this constraint, the tonic baseline activity could be removed before dimensionality reduction. Consequently, resulting synergies could indicate both mutual excitation and reciprocal inhibition of muscles (based on the sign of their respective weighting factors), and their activation was interpreted with respect to baseline activity levels.

Statistics

The time window on which the statistical analysis was performed spanned the first 1.25 s after the perturbation. This encompassed the entire movement of the platform as well as the initial and corrective responses reflected in the CoM acceleration and rate of change in angular momentum. One-way repeated-measures ANOVAs were used to identify differences between sessions (Pre, Post1, Post2) on all synergy and kinematic measures. Statistical Parametric Mapping (SPM) was used to perform statistical analysis on time series level (Friston et al., 2006; Pataky, Robinson & Vanrenterghem, 2016). When a seemingly single effect was split into multiple significant clusters a combined p-value, denoted by p*, was obtained using Fisher’s method (Fisher, 1936). In case of significant differences between sessions, post-hoc two-tailed repeated-measures t-tests were performed between sessions (Pre-Post1, Pre-Post2, Post1-Post2) with Bonferroni correction. For all statistical inferences an α-level of 0.05 was used.

Results

Recovery responses

Kinematics

The mean CoM displacement, velocity, and acceleration, as well as orientation, angular momentum, and rate of change of angular momentum are displayed in Fig. 1 (lateral perturbations) and Fig. 2 (medial perturbations), along with the platform’s angle of rotation for reference. Both linear and rotational parameters are the result of gravity and a single external force applied on the body, the GRF. The latter force, of course, changes due to the platform rotation. The resulting perturbation can only be countered by modulating the orientation and magnitude of the GRF vector (through horizontal GRF components) and its point of application (CoP).

Figure 1 Kinematic response to lateral perturbations.

Group average linear kinematics (A, C, E) and rotational kinematics (B, D, F) for a [−0.5, 2.5] s window around onset of lateral perturbations. The red lines represent kinematic responses; the grey line represents the rotation angle of the platform and is scaled per graph. Line types reflect sessions: Pre (— solid line), Post1 (••• dotted line), and Post2 (–-– dashed line). The grey area indicates the time window that is analysed. The illustration (G) depicts the positive linear and rotational directions (green) and the direction of the platform rotation (red) for reference.

Figure 2 Kinematic response to medial perturbations.

Group average linear kinematics (A, C, E) and rotational kinematics (B, D, F) for a [−0.5, 2.5] s window around onset of medial perturbations. The black lines represent kinematic responses; the grey line represents the rotation angle of the platform and is scaled per graph. Line types reflect sessions: Pre (— solid line), Post1 (••• dotted line), and Post2 (–-– dashed line). The grey area indicates the time window that is analysed. The illustration (G) depicts the positive linear and rotational directions (green) and the direction of the platform rotation (red) for reference.

By and large, both the linear and rotational kinematic parameters displayed the same behaviour after perturbations for the two perturbation directions, but in opposite directions. In the initial 0.4 s the CoM accelerated as a result of the platform rotation (Figs. 1E and 2E). The CoM acceleration revealed a corrective response that took effect approximately 0.4 s after the perturbation and lasted until almost 1 s after the perturbation (Figs. 1E and 2E). During this period the change in angular momentum (angular momentum strategy) coincided with the perturbation direction (Figs. 1F and 2F), contributing to the corrective CoM acceleration. As the corrective CoM acceleration was not fully accounted for by the rate of change in angular momentum, a CoP shift (ankle strategy) in the perturbation direction also contributed to the correction of the CoM.

The angular momentum changes in response to the perturbation and the platform’s rotation back to its neutral orientation led to a substantial overshoot in orientation within 1 s after perturbation onset. The overshoot in CoM position, however, was only moderate (lateral perturbation) to small (medial perturbation) and occurred 1.5 and 2 s after perturbation onset, respectively.

Muscle activity

The activity of all muscles was decomposed into five synergies (Fig. 3). These five synergies explained 70.5% of the total variance. Most of the muscle activation was captured in synergies 1 and 2 (accounting for 32.8% and 20.0% of the total variance, respectively), while synergies 3 (8.9%), 4 (7.5%), and 5 (4.9%) contributed considerably less. Synergy 1 appeared to act to increase leg stiffness through co-contraction around the ankle (TAD-PLD), knee (VLD/RFD-BFD), and hip (RFD-BFD) joints. Activation of this synergy rose shortly after the perturbation, regardless of its direction, and remained high throughout balance recovery. This synergy appeared to be more active after medial than lateral perturbations. Synergy 2 acted to generate torque around the ankle by having opposing effects on the TAD and PLD. Befitting this function, its inhibiting and exciting effects were opposed between perturbation directions. Its activation seemed to be involved in CoM acceleration given their resembling patterns over time (Figs. 1E and 2E). Its initial activation resembles a stretch reflex (reciprocal inhibition). This was quickly followed by a response that would induce a shift in CoP towards the perturbation direction, a response that was inferred from the kinematics as well. Synergies 3, 4, and 5 individually have small contributions to the overall response and their main muscles already have a function in the two principal synergies, making it increasingly difficult to ascertain their particular function.

Figure 3 Muscle synergy response to perturbations.

Muscle weighting factors (left) and group average synergy activation profiles (right) after lateral (red) and medial (black) perturbations. Line types reflect sessions: Pre (— solid line), Post1 (••• dotted line), and Post2 (--- dashed line). The grey area indicates the time window that is analysed.

Effects of training

Lateral perturbations

The responses to lateral perturbations of the primary parameters of interest and their SPM F-statistics are shown in Fig. 4. The other parameters and the post-hoc t-tests can be found in Material S3. The variability of the parameters within Pre, Post1, and Post2 is illustrated in Material S5 and the variability in activation of the individual muscles can be found in Material S7. The training programme did not alter CoM displacement nor its velocity directly, but it reduced the CoM acceleration in the later phase of the corrective response (p = 0.015) as well as the subsequent deceleration (p = 0.007) (Figs. 4A–4D). Post-hoc analysis indicated that the former effect was only significant after the full training programme (Pre-Post: p = 0.012). The total body orientation was not significantly affected by training. However, the corrective angular momentum response, to regain upright body position but ended in overcorrection, was reduced after training (ANOVA: p < 0.001; Pre-Post1: p = 0.007; Pre-Post2: p = 0.005) (Fig. 4B). This was achieved by shortening the duration in which the rate of change in angular momentum was positive (p* = 0.007) (Fig. 4E). Post-hoc analysis showed no significant effects, indicating a training effect that was too subtle to differentiate between individual session pairs. Also the subsequent negative rate of change in angular momentum response was reduced after training (ANOVA: p < 10−6, Pre-Post1: p = 0.001, Pre-Post2: p < 10−5), limiting the overshoot in body orientation. These effects appear to be responsible for the observed adaptations in CoM acceleration. The activation of synergy 1 was reduced only after the first training session (ANOVA: p* < 0.009; Pre-Post1: p* < 10−4), but not after the full training program (Fig. 4C). No changes were observed in the other four synergies.

Figure 4 Statistical analysis lateral perturbations.

Rows 1 & 3: the group average linear kinematics (left column), rotational kinematics (middle column), and synergy activation profiles (right column) after lateral perturbations. The red lines represent observed responses; line types reflect sessions: Pre (— solid line), Post1 (••• dotted line), and Post2 (–-– dashed line). Rows 2 & 4: the SPM F-statistic in time belonging to the panel above it. The red dotted line in these graphs represent the ANOVA’s critical F-value, values above this line are considered significant. The grey areas indicate the time window that is analysed.

Medial perturbations

The responses to medial perturbations of the parameters of primary interest and their SPM F-statistics are shown in Fig. 5. The other parameters and the post-hoc t-tests can be found in Material S4. The variability of the parameters within Pre, Post1, and Post2 is illustrated in Material S6 and the variability in activation of the individual muscles can be found in Material S8. No significant changes in linear kinematics were observed in the early recovery response. Also the total orientation of the body after perturbation was unaffected by training. The initial induced angular momentum was smaller and counteracted sooner after training (p = 0.041) (Fig. 5B). Post-hoc t-tests between the Pre and other sessions were non-significant for both contrasts, indicating that the overall training effect was too subtle to differentiate between individual pairs of sessions. A smaller induced angular momentum would require a smaller corrective response. This was also observed (p = 0.005), but only after the full training program (Pre-Post2: p < 0.001) (Fig. 5B). These effects were realised by a shorter corrective rate of change in angular momentum response (p = 0.003) (Fig. 5E), again only after the full training program (Pre-Post2: p = 0.003). The secondary rate of change in angular momentum response was reduced by training as well (p = 0.042), although post-hoc t-tests indicated this effect was not particular to either of the two Post sessions. Notably, neither effect on rate of change in angular momentum was reflected in the CoM acceleration. Activation of synergy 1, involved in leg stiffness, was reduced by training (p = 0.049) (Fig. 5C). This effect was significant for the Pre-Post1 contrast (p < 0.001). Activation of synergy 2, involved in ankle torque, was only affected in its initial response, i.e., the presumed stretch reflex (p = 0.044) (Fig. 5F). Post-hoc analysis indicated this quicker repression after training was not significant between any particular sessions. No changes were observed in the other three synergies.

Figure 5 Statistical analysis medial perturbations.

Rows 1 & 3: the group average linear kinematics (left column), rotational kinematics (middle column), and synergy activation profiles (right column) after medial perturbations. The black lines represent observed responses; line types reflect sessions: Pre (— solid line), Post1 (••• dotted line), and Post2 (–-– dashed line). Rows 2 & 4: the SPM F-statistic in time belonging to the panel above it. The red dotted line in these graphs represent the ANOVA’s critical F-value, values above this line are considered significant. The grey areas indicate the time window that is analysed.

Discussion

Aging degrades the ability to recover from balance perturbations, but balance training programs offer a means to partially counteract these effects (Sherrington et al., 2017, 2019). Understanding the mechanisms through which training enhances balance control is paramount in refining such programs and tailoring them to individual needs. In a previous study, we highlighted the positive impact of training on balance performance and increased ankle muscle co-contraction during both unperturbed and perturbed unipedal balancing (Alizadehsaravi et al., 2022b). We suggested that this increased ankle joint stiffness might compensate for the age-related deficits in sensorimotor control and, as such, reflect improved feedforward balance control. However, the significance of feedback control, particularly in response to more sudden and unexpected perturbations, cannot be overlooked. Moreover, it has been demonstrated that the use of feedback control movement strategies in balancing is substantially influenced by aging (Afschrift et al., 2016; Alcock, O’Brien & Vanicek, 2018; Amiridis, Hatzitaki & Arabatzi, 2003; Manchester et al., 1989). In the present study, participants experienced mediolateral perturbations before and after a balance training program. The random timing and direction of the perturbations limited the potential for anticipatory control. Consistent responses in muscle activation and kinematics were observed after perturbation, indicating reliance on feedback control. Balance training induced changes in these responses, leading to improved balance recovery.

Balance recovery

While the ankle strategy appears to be dominant in less challenging conditions (Creath et al., 2005; Patel et al., 2008) it has been suggested that the angular momentum strategy may be employed to control balance when actions exerted through ankle torques are inadequate (van Dieën, van Leeuwen & Faber, 2015). The perturbations in the current experiment were sufficiently challenging; in about a third of the trials participants made minor touches to the handrail. Hence, use of ankle and angular momentum strategies was to be expected. This was indeed observed in the corrective response. At perturbation onset medial and lateral platform rotation induced corresponding medial and lateral accelerations of the CoM. Concurrently, translation of the ankle joint (through rotation of the platform and standing foot) and inertia of the body induced a change in angular momentum opposing the perturbation direction. Subsequently, a corrective response was initiated. Joint stiffness was increased (see synergy 1) and an ankle torque was generated to counteract the imposed CoM acceleration (see synergy 2). In addition to the ankle strategy, an angular momentum was generated that contributed to the corrective CoM acceleration, but also rotated the body past the vertical in the direction of the perturbation. Interestingly, when the CoM velocity reversed, the two strategies started to be used differently. The ankle strategy continued to accelerate the CoM in corrective direction, while the rate of change in angular momentum became corrective for the body’s orientation instead, at the cost of accelerating the CoM in the direction of the perturbation.

The muscle activation underlying the balance recovery response could for a large part be explained by just two muscle synergies. These appeared to represent distinct aspects of balance control. Exemplified by the activation irrespective of perturbation direction, synergy 1 was not used to produce torques around specific joints. Instead, it represents co-contraction of antagonistic muscles over the ankle, knee, and hip joints, presumably increasing leg stiffness. Although co-contraction has also been suggested as evidence of feedforward control, the temporal activation pattern of this synergy showed its onset was a response to the perturbation. Furthermore, it should be noted that EMG data were baseline corrected, so these results are naïve of (feedforward) co-contraction activation present before the perturbation. Hence, this co-contraction is interpreted as a feedback control component. Synergy 2 consisted primarily of reciprocal inhibition of the tibialis anterior and peroneus longus. As it served to produce torque around the ankle it contributed to the ankle strategy. Besides its contribution to this feedback control strategy, synergy 2 may potentially also encompass the stretch reflex, which aggravated the effect of the perturbation. This would then be represented by the initial response, which peaked when the platform rotation, and therefore muscle stretch, was at its fastest. It should be noted that caution is generally warranted when interpreting negative weighting factors in synergy analysis as inhibition (Tresch, Cheung & D’Avella, 2006). However, the data observed here presented with substantial muscle tonus, limiting instances of indeterminable inhibition of muscle activation that would invalidate such an interpretation.

Perturbation direction

CoM acceleration was only affected by training for perturbations in the lateral direction. Possibly, unipedal balance control prioritises control of the CoM after perturbations in the lateral direction more than after those in the medial direction. Generally perturbations in the medial direction can easily be resolved by the change in support strategy (putting the other foot down), whereas perturbations in the lateral direction have a much greater fall risk. This aligns with the apparently smaller maximum CoM displacement, ‘ankle torque’-synergy activity, and initially induced angular momentum for lateral compared to medial perturbations. Additionally, it would explain the larger overshoot in CoM displacement in the safer medial direction (after lateral perturbations) than in the risky lateral direction (after medial perturbation), as well as the faster CoM recovery from lateral than medial perturbations. Alternatively, changes in angular momentum responses to medial and lateral perturbations may arise from a limited range of motion of the ankle joint to cope with medial perturbations (Menadue et al., 2006), which is exacerbated with age (Nigg et al., 1992). The inability to rotate the ankle to accommodate platform rotation would place a constraint on the orientation of proximal segments. Moreover, the ankle reaching extreme joint angles sooner in this direction may account for the larger response in synergy 2 (Burgess et al., 1982; Burgess & Clark, 1969).

Training effects

Balance control improved after training (Alizadehsaravi et al., 2022b), as reflected by changes in linear and rotational kinematics. These changes can be attributed to improvements in feedback control; they were mostly observed in the reactive response, whereas feedforward control would also affect the initial, induced, kinematics. The effect of balance training on feedback control after perturbations was primarily observed in rotational kinematics after the 3-week training program. For recovery from lateral perturbations these effects were already present after a 45-min training session. Note that such rapid improvements are too fast for physiological adaptations in the muscles (Abe et al., 2000; DeFreitas et al., 2011; Monti et al., 2020), indicating that they are a product of changes in sensorimotor processing instead. This is further corroborated by the decrease in activation of synergy 1 for medial and lateral perturbation recovery after only 45 min of training. Though ostensibly transient, this quick neural adaptation of decreasing synergy 1 activity may already be beneficial: it reduces joint stiffness, negating the effect of platform rotation that leads to the corrective overshoot.

Training decreased angular momentum and its rate of change during balance recovery from medial and lateral perturbations. Hence, use of the angular momentum strategy to correct CoM displacement was reduced after training. While limiting the overshoot in body orientation, this led to a reduced corrective CoM acceleration for lateral perturbations. However, for medial perturbations, more ankle strategy was used to compensate. From this we infer that training enhances feedback control strategies. Since the intensity of the feedback response was not just scaled in its entirety, but rather the two control strategies were tuned separately to achieve a common goal, we speculate it is the sensorimotor processing underlying the individual feedback control loops that is enhanced. This is further supported by the dissimilar training adaptations in feedback control strategies for medial and lateral perturbations.

For recovery from medial perturbations we also observed evidence for improvements in feedforward control after training. Although it could not be attributed to a single session contrast, training appeared to decrease the induced angular momentum after medial perturbations. It also appeared to shorten the initial response of synergy 2, the potential stretch reflex. As such, attenuation of the reflex gain may have reduced the impact of the platform rotation allowing for faster development of an ankle torque in the opposite direction. This could explain the smaller effect of the perturbation on angular momentum which only occurred after medial perturbations. However, it should be noted that in our previous study adaptation of reflex gains, at least in the form of Hoffmann reflex in the soleus, was not found (Alizadehsaravi et al., 2022b). This suggests that modulation of the gain was not due to pre-synaptic inhibition but due to a reduced gamma drive. Alternatively, the observed effect may not be reflex modulation at all, but rather a quicker corrective response.

Since there was no control group, natural variability over time within subjects could not be assessed. However, the inclusion of two post-training measurements (short-term & long-term training) enhances the reliability of our findings. If the observed effects were due solely to natural variability, we would expect random fluctuations over contrasts. Instead, the observed effects were generally consistent in direction, even though not always statistically significant. This consistency suggests that the observed effects are indeed attributable to the intervention.

Despite potential changes in feedforward control we found that adaptations in feedback control contributed to improvements in balance performance after training. Since these adaptations were situation and strategy specific, we propose that attunement of the ankle and angular momentum strategies is the result of improved sensorimotor processing. Note that this improvement in processing may affect automated as well as volitional responses. Furthermore, as can be expected from feedback control, it appears the refinement in balance recovery after training was not achieved through faster responses but rather through fine-tuning of the later responses to reduce corrective overshoot, particularly after lateral perturbations.

Conclusions

We investigated the impact of balance training in older adults on feedback control following unpredictable unipedal balance perturbations, induced by rotations of the support surface. Training improved balance recovery primarily through fine-tuning of feedback control strategies, leading to better tuned corrective responses after a perturbation. The improvement was primarily observed in reduced angular momentum and its rate of change, but also centre of mass acceleration and synergistic muscle activation leading to leg stiffness and ankle torques were reduced. This mainly served to attenuate the corrective overshoot, rather than respond quicker. The changes in feedback control strategies are suggested to be the result of improved sensorimotor processing.

Supplemental Information

Supplemental Information 1 Supplementary material.

Additional Information and Declarations

Competing Interests

Author Contributions

Human Ethics

Data Availability

Jaap van Dieen is an Academic Editor for PeerJ.

Ruud A. J. Koster conceived and designed the experiments, performed the experiments, analyzed the data, prepared figures and/or tables, authored or reviewed drafts of the article, and approved the final draft.

Leila Alizadehsaravi conceived and designed the experiments, performed the experiments, analyzed the data, prepared figures and/or tables, authored or reviewed drafts of the article, and approved the final draft.

Wouter Muijres conceived and designed the experiments, performed the experiments, authored or reviewed drafts of the article, and approved the final draft.

Sjoerd M. Bruijn conceived and designed the experiments, analyzed the data, authored or reviewed drafts of the article, and approved the final draft.

Nadia Dominici analyzed the data, authored or reviewed drafts of the article, and approved the final draft.

Jaap H. van Dieën conceived and designed the experiments, analyzed the data, authored or reviewed drafts of the article, and approved the final draft.

The following information was supplied relating to ethical approvals (i.e., approving body and any reference numbers):

The review board of the Faculty of Behavioural and Movement Sciences of VU Amsterdam granted approval to carry out the study (VCWE-2018-171).

The following information was supplied regarding data availability:

The data and code (Matlab) are available at Zenodo: Koster, R., Alizadehsaravi, L., Muijres, W., Dominici, N., Bruijn, S., & van Dieen, J. (2024). Koster et al. - Balance training in older adults enhances feedback control after perturbations—data & code [Data set]. Zenodo. https://doi.org/10.5281/zenodo.14013823.

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
