# Peer review of "Balance training in older adults enhances feedback control after perturbations"

_PeerJ, doi:10.7717/peerj.18588_

## Round 0.1 · original submission · Major Revisions

The reviewers request greater clarification about your study purpose and the design implemented to address that purpose. The introduction also needs to be revised in order to set up this specific studies purpose.

·

Basic reporting

1. As stated, they hypothesis is not directly testable. The proposed study does not seem designed to test for changes in sensorimotor processing. Changes in sensorimotor processing could certainly be an interpretation of the results, but it does not seem suitable for the hypothesis. Instead, consider specific expected outcomes from the training on your primary dependent variables.

2. It would be helpful in the introduction if the authors could cite evidence that one of feedback or feedforward control is more impaired in older adults?

3. To account for reach-to-grasp as well as stepping, consider the term “change in support” strategy rather than “stepping” strategy in the introduction.

3. It was not totally clear why the authors would expect changes in feedback control given the content of the training program. Additional details to strength the motivation and justification for this would be helpful.

Experimental design

1. Several studies and reviews have outlined the limitations and drawbacks of using PCA for muscle synergy extraction from EMG data (e.g., lacks physiological interpretation). Traditionally non-negative matrix factorization has been used for muscle synergy analysis. Additional justification for the use of PCA rather than NNMF, and how any limitations of PCA (methodological and interpretative) were overcome is required.

2. Compared to previous muscle synergy research, 70% variance accounted for seems like a low threshold for selecting the number of muscle synergies. Historically it has been closer to 90%. What was the justification for such a departure from the literature? To me, this suggests that there may be insufficient variance in the EMG data to make decomposition useful.

3. It was not immediately clear why a unipedal stance was selected. Please clarify and justify this choice.

4. Whole body angular momentum was calculated about an axis passing through the whole body center of mass. Several publications have made appealing biomechanical arguments that whole body angular momentum should be calculated using the axis of rotation about which the perturbation / balance challenge is occurring. For example, see:

Chiovetto et al., (2018). Low-dimensional organization of angular momentum during walking on a narrow beam.

Additional justification for calculation of angular momentum about the COM rather than the axis of the platform is required.

5. Were any checks used to ensure that there were no substantial feedforward or anticipatory changes by participants (e.g., posture, weight bearing asymmetry)?

Validity of the findings

1. It is not immediately clear to me whether participants recovered or maintained their balance. Based on the presentation of the results it initially appeared as though all participants maintained rather than recovered their balance as they did not have to use a change-in-support strategy (step or reach). However, in the discussion it was mentioned that a number of trials resulted in participants grabbing a bar nearby. How many of those trials were there? In how many participants? Were those trials retained in the analysis or removed? Consideration for how these data were handled and whether the appropriate terminology in the discussion is “recovered balance” or “maintained balance” is needed.

2. To avoid confusion, remove reference to results as “trending”. Results should be presented clearly as significant or not significantly different.

3. Lines337-340: This discussion point seems overly speculative. Recommend removing it.

Reviewer 2 ·

Basic reporting

Koster et al. assessed the effect of balance training in older adults using kinematic and electromyographic (EMG) data collected during balance recovery responses to unpredictable mediolateral perturbations. They observed an improvement in balance performance characterized by a reduced center of mass (CoM) acceleration and angular momentum that the authors attributed to an improved sensorimotor processing as suggested by a decreased muscle activation. Despite the study design and experimental data might be appropriate to answer the research question, I have several concerns about the analysis and interpretation of the results. In particular, rather than using global parameters (such as CoM, angular moment, and muscle synergies) the work would benefit from a deeper analysis of individual muscles and segments.

Experimental design

The rationale of using the synergy analysis approach is unclear. While muscle synergies might give a general description of the control strategies adopted by the participants, an individual muscle analysis would be more appropriate to answer a question related to changes in feedback/feedforward responses to perturbations.
The participants were allowed to use their arms to counteract the perturbation. The use of the arms during a balance task might change completely the body reactive response. However, no EMGs were placed on the arms, and the analysis of the kinematics only considered global variables such as the center of mass and angular moment. I suggest to add the analysis of individual segments.

Validity of the findings

The changes in initial muscle responses seem too late to be interpreted as changes in stretch reflexes (>250ms). Whether the improvement in balance recovery can be associated to an improved sensorimotor processing or a better volitional strategy cannot be answered without a deeper analysis of individual muscle responses and neuromechanical delays.

Additional comments

The introduction needs to be revised. In particular there is only a brief mention to balance training, making it difficult to frame the work in the context of the gap in literature concerning the effect of balance training.
Did the participant had a familiarization procedure with the balance perturbations? Differences across sessions may also arise from learning a better strategy to recover balance rather than be the effect of training. Having a control group could also help to address this limitation.
Line 107: I suggest to change the words “presumed neurophysiological correlates”. These terms are mostly used for cortical analysis and might be misleading in this context.
Line 133-135: Participants were allowed to take safety countermeasures to avoid falls (for example foot down or grabbing the handrails). Were those trials excluded from the analysis?
It is unclear whether the figures refers to individual or group means. In the latter case, I strongly suggest to add individual means to give an idea of the variability in control strategies across subjects.

---

## Round 0.2 · Minor Revisions

Both reviewers found merit in your manuscript and thought you addressed the majority of concerns. Reviewer 2 has made a recommendation that may help solidity your conclusions.

·

Basic reporting

All questions and comments have been addressed.

Experimental design

No comment

Validity of the findings

No comment

Additional comments

No comment

Reviewer 2 ·

Basic reporting

The authors have addressed my concerns in a clear and meaningful way. I still honestly have difficulty in understanding the use of synergy analysis to answer the research question. Synergy analysis would need enough variance in the dataset, otherwise would just show patterns of coactivation. Indeed, the synergies found in this study are quite different from previous works (Chvatal et al., 2011; Chvatal & Ting, 2013; Torres-Oviedo & Ting, 2007, 2010). I understand that individual muscle analysis level would substantially increase the number of comparisons, but the analysis of at least a subset of muscles would corroborate the findings. Furthermore, caution should be used in interpreting negative values as inhibitory, particularly with PCA (see Tresch MC et al. Matrix factorization algorithms for the identification of muscle synergies: evaluation on simulated and experimental data sets. J Neurophysiol. 2006).

Experimental design

'no comment'

Validity of the findings

'no comment'

Additional comments

'no comment'

---

## Round 0.3 · accepted · Accept

The reviewers were pleased with the authors corrections. The manuscript is ready for publication.

Reviewer 2 ·

Basic reporting

no comment

Experimental design

no comment

Validity of the findings

no comment